# A Comparative Density Functional Theory Study of Hydrogen Storage in Cellulose and Chitosan Functionalized by Transition Metals (Ti, Mg, and Nb)

**DOI:** 10.3390/ma15217573

**Published:** 2022-10-28

**Authors:** Omar Faye, Jerzy A. Szpunar, Ubong Eduok

**Affiliations:** Department of Mechanical Engineering, College of Engineering, University of Saskatchewan, 57 Campus Drive, Saskatoon, SK S7N 5A9, Canada

**Keywords:** cellulose, chitosan, hydrogen storage, magnesium, titanium, niobium

## Abstract

The focus of this work is hydrogen storage in pristine cellulose, chitosan, and cellulose. Chitosan doped with magnesium, titanium, and niobium is analyzed using spin unrestricted plane-wave density functional theory implemented in the Dmol^3^ module. The results of this study demonstrate that hydrogen interaction with pure cellulose and chitosan occurred in the gas phase, with an adsorption energy of $E_{b}$ = 0.095 eV and 0.090 eV for cellulose and chitosan, respectively. Additionally, their chemical stability was determined as $E_{b}$= 4.63 eV and $E_{b}$ = 4.720 eV for pure cellulose and chitosan, respectively, by evaluating their band gap. Furthermore, the presence of magnesium, titanium, and niobium on cellulose and chitosan implied the transfer of an electron from metal to cellulose and chitosan. Moreover, our calculations predict that cellulose doped with niobium is the most favorable medium where 6H_2_ molecules are stored compared with molecules stored in niobium-doped chitosan with T_max_ = 818 K to release all H_2_ molecules. Furthermore, our findings showed that titanium-doped cellulose has a storage capacity of five H_2_ molecules, compared to a storage capacity of four H_2_ molecules in titanium-doped chitosan. However, magnesium-doped cellulose and chitosan have insufficient hydrogen storage capacity, with only two H_2_ molecules physisorbed in the gas phase. These results suggest that niobium-doped cellulose and chitosan may play a crucial role in the search for efficient and inexpensive hydrogen storage media.

## 1. Introduction

The increase in greenhouse gas emissions in the atmosphere accelerates global warming that threatens life on earth [1].Therefore, a smooth transition from fossil fuels to clean and sustainable fuels is required. Owing to its high energy content per mass (142 KJ/g) compared to petroleum (47 KJ/g), hydrogen is an optimum clean fuel; when used in fuel cells to generate electricity, it produces water [2,3]. However, the use of H_2_ in the future clean economy will still face many obstacles and require various problems to be addressed. When generated, hydrogen has to be stored, and finding a safe and efficient hydrogen storage medium is important for the future hydrogen-based economy. Many hydrogen storage methods have been proposed to date. The most well-known method involves storing gas H_2_ in high-pressure tanks [4,5]. However, compressing gaseous hydrogen in the pressure range of 70–80 MPa in a tank creates an additional risk of hydrogen-related embrittlement of the tank walls. Hydrogen can also be stored in cryogenic tanks, although this method requires thermal insulation and cooling of hydrogen below the critical temperature of 33 K. Significant energy is required to liquify H_2_, coupled with continuous boil-off. Therefore, to find an alternative way of storing hydrogen, the U.S. Department of Energy (DOE) initiated support for the extensive search for the materials for H_2_ storage media in 2003. The criteria set by the DOE for materials for onboard hydrogen storage were specified as (i) high storage capability, that is, 5.5 wt% and 40 g/L at ambient temperatures; (ii) rapid H_2_ release and recharge under moderate conditions; and (iii) long recycling life, that is, more than 1000 recharge and discharge cycles [6]. However, reaching a high storage capacity of 5.5 wt% and 40 g/L under ambient conditions, fast H_2_ release, and recharge under moderate conditions is a difficult task. Therefore, to address these challenges, the DEO set an ultimate goal of 6.5 wt% and 60 g/L by 2050 [3]. The following additional criteria were set: an operating temperature (℃)in the range of [−40, 60], a minimum and maximum delivery temperature of −40 and 85 ℃, respectively, 1500 operational cycles, delivery pressure in the range [5,6] (bar), and an onboard efficiency of 90% [3]. Furthermore, the rate of charging and discharging was specified as 3–5 min, with a minimum cost of USD 266/kg H_2_ [3]. Many materials have been examined and tested to date. There are three ways that hydrogen can interact with the material: physisorption, chemisorption, and the quasi-molecular form of Kubas interaction [2]. During physisorption, hydrogen remains in the molecular form and binds weakly on the substrate, with binding energy in the meV range. This weak binding implies that low temperature is sufficient to desorb the H_2_. Such behavior is observed in the metal–organic framework [7,8,9,10,11,12,13,14,15] and in carbon-based materials [16,17,18,19]. However, during chemisorption, hydrogen molecules dissociate into atoms, and the atoms diffuse into the substrate and bind chemically with enthalpy energy in the range of 2 to 4 eV [4]. This strong bonding makes hydrogen desorption difficult, and the process takes place only at high temperatures. This type of bonding occurs in light metal hydrides and chemical hydrides [20,21]. During the Kubas interaction, H_2_ bonds stretch without braking. The bonding strength is between that physisorption and chemisorption in the range of [0.1–0.8 eV]. Such bonding occurs in nanostructure materials and functionalized sorbent materials [20]. There are other requirements for material performance, including the hydrogen storage medium. The material should also be non-toxic, environmentally friendly, and widely available. Biopolymers, such as cellulose and chitosan, satisfy such requirements and show potential for hydrogen storage. Cellulose and chitosan are used in wastewater treatment. They are suitable sorbents for heavy metals, dyes, and organic and emerging contaminants [22,23]. Udoetok et al. [24] proposed chitosan/cellulose glutaraldehyde composite materials synthesized with variable morphology and surface properties with varying levels of self-assembly and cross linking. Their study revealed the structural and synergistic effects of the adsorption properties of cellulose–chitosan composites, as well as their potential application for advanced water treatment, nanomedicine, and drug delivery [24]. Cellulose has attracted attention because it is recyclable and abundant. Furthermore, it can also be used for electronic component applications and in energy devices [25,26]. Mokena et al. [27] summarized cellulose applications in sensors, as well as biomedical, wastewater treatment, and packaging industries. Cellulose combined with graphene has a variety energy storage, electronic, biomedical, optical, and catalysis applications [28]. In addition to cellulose, chitosan also can be used in biomedical [29,30], water treatment [31,32], and environmental applications [33]. However, to the best of our knowledge, no study has been conducted on hydrogen storage in cellulose and chitosan functionalized with magnesium, titanium, and niobium. Therefore, highlighting the effectiveness of these composites for hydrogen storage is important in the ongoing search for materials. In the present study, a density functional theory is used to investigate hydrogen storage in pure cellulose and chitosan. The effect of magnesium-, titanium-, and niobium-coated cellulose and chitosan is also tested to evaluate hydrogen storage characteristics.

## 2. Computational Methods

We performed a first-principles calculation using spin unrestricted plane-wave density functional theory with the self-consistent field method implemented in the Dmol^3^ module [34,35]. We used generalized gradient approximation (GGA) with Perdew–Burke–Ernzerhof (PBE) [36] to approximate the exchange-correlation effects on electron−electron interactions. The semi-core-pseudopotentials represent the core electrons as a single effective potential [37]. Double-numerical plus polarization (DNP) used as a basis set, included the vdW interaction (DFT-D), as proposed by Grimme [38]. Energy minimization was achieved with a convergence tolerance energy of 10^−5^ Ha. The atomic positions were relaxed such that the force acting on each atom was less than 0.002 Ha/Å.

## 3. Results and Discussion

We investigated the interaction between pure cellulose and transition metals (T.M.s), such as titanium (Ti), magnesium (Mg), and niobium (Nb). Similar calculations were performed for chitosan with the same metals. The strength of the interaction was measured using the following equation:(1)$$E_{b}=E_{\left(A\right)}+E_{\left(B\right)}-E_{\left(AB\right)}$$
where $E_{\left(A\right)}$ is the total energy of transition metals in an isolated cubic cell of lattice (a = b = c = 25 Å), $E_{\left(B\right)}$ is the ground energy of cellulose or chitosan, and $E_{\left(AB\right)}$ is the total energy of cellulose or chitosan doped with the mentioned TMs. The optimized structures of cellulose and chitosan are displayed in Figure 1.

We also studied the frontier molecular orbitals to gain insight into the interaction of pristine cellulose and chitosan with additives such as niobium, titanium, and magnesium. Figure 2 depicts the highest occupied molecular orbital (HOMO) (Figure 2A) and the lowest unoccupied molecular orbital (LUMO) (Figure 2B) for pristine cellulose. The same analysis was performed on pristine chitosan, the highest occupied molecular orbital (HOMO) (Figure 2C) and the lowest unoccupied molecular orbital (LUMO) (Figure 2D) or which are also plotted in Figure 2. Furthermore, the energy band gap was determined according to Equation (2).
(2)$$E_{g}=E_{LUMO}-E_{HOMO}$$
where $\;E_{LUMO}$ is the energy of the lowest unoccupied molecular orbital (LUMO), and $E_{HOMO}$ is the energy of the highest occupied molecular orbital (HOMO).

The energy difference between HOMO and LUMO orbitals determines the chemical stability of a molecule. These molecular frontiers were calculated using Dmol^3^ implemented in the material studio.

In Equation (2), a low $E_{g}$ value indicates the ability to donate electrons to the additive atoms.

Furthermore, the global chemical activity, hardness, and softness parameters were investigated. The ionization energy is defined as I = −$E_{HOMO}$, which is the minimum energy required to remove an electron from a molecule in the gas phase. The electron affinity is defined as A = −$E_{LUMO}$, which is the energy increase that occurs when an electron is added to a molecule in the gas phase. The chemical hardness (α) is α = (I − A)/2, measuring the inhibition activity of charge transfer within the molecule. The chemical softness is represented by S = 1/2α. In addition, Mulliken electronegativity is defined as (I + A)/2, which represents the ability of an atom in a molecule to attract electrons. Finally, the chemical potential and the maximum charge transfer parameter are defined as µ= − (I + A)/2 and Dn = (I + A)/2(I − A), respectively. These chemical parameters are summarized in Table 1.

The HOMO and LUMO energy levels are −16.899 eV and −12.269 eV, respectively, for pristine cellulose compared to −5.918 eV and −1.196 eV for chitosan, as displayed in Table 1. Moreover, the ionization energy, electronegativity, chemical potential, chemical hardness, and chemical softness ranging between −14.584 eV and 16.899 eV for cellulose compared to −3.557 eV to 5.918 eV for chitosan, as shown in Table 1. The maximum charge transfer (Dn) is 3.149 eV and 1.507 eV for cellulose and chitosan, respectively. These results indicate that cellulose and chitosan are highly stable. Because cellulose presents with six carbon asymmetric and three oxygen atoms in a different state, it is important to search for the most favorable adsorption site of magnesium, titanium, and niobium on cellulose. Equation (1) shows that the most favorable site is the oxygen in the bridge position, as shown in Figure 3, with an adsorption energy of 1.890 eV, 3.720 eV, and 3.726 eV for Mg, Ti, and Nb, respectively. A similar investigation was performed on chitosan, with five asymmetric carbons, four oxygens in different states, and one nitrogen type, as shown in Figure 1. The favorable adsorption site of magnesium and titanium is the O_Bridge position, with a binding energy of 1.547 eV and 5.450 eV for Mg and Ti, respectively. However, for the niobium atom, the most favorable site is the N_Top site, as shown in Figure 3, with a binding energy equal to 8.185 eV (Table 2). These high symmetry points are displayed in Figure 3.

The calculated enthalpy energy is summarized in Table 2. The interaction of pristine cellulose with titanium is the most stable, with a binding energy of $E_{bTC}=$3.720 eV and an optimized distance of d_CelTi = 2.340 Å, followed by interaction between cellulose and niobium, with $E_{bNbCel}=3.572\;\mathrm{eV}$ and a final distance of d_CelNb = 2.312 Å. The lowest binding energy is associated with the interaction of cellulose with magnesium, with $E_{bMg}=1.890\;\mathrm{eV}$ and a minimum distance of d_CelMg = 2.35 Å. A comparative study was also performed on chitosan, for which the most favorable interaction occurs between chitosan and niobium, with an adsorption energy of $E_{b}Nbch\;$= 7.180 eV and a critical distance of d_ChNb = 2.110 Å for the nearest atom of chitosan. The enthalpy reaction of titanium with chitosan is E_b_Tich = 5.450 eV, with a required length of 3.225 Å compared to 3.720 eV in the case of titanium-doped cellulose (d_CelTi= 2.32 Å). Finally, magnesium adsorption energy on chitosan is 1.547 eV, with an adsorption distance of 3.666 Å compared to 1.890 eV and d_CelMg = 2.544 Å in the case of magnesium-doped cellulose. Binding energy and bandgap energy were computed using Equations (1) and (2). These results are summarized in Table 2.

To investigate the binding process between pure cellulose or chitosan with magnesium, titanium, and niobium, we studied the fluctuation of cellulose’s bandgap energy and chitosan’s bandgap energy in the presence of the mentioned transition metals. We defined the bandgap of the investigated materials using Equation (2).

The calculated values of the bandgap energy are presented in Table 2. Analysis of the results reveals that the bandgaps of pure cellulose and chitosan are 4.630 and 4.720 eV, respectively, in agreement with results reported in the literature [27,39,40,41].

When we doped cellulose with magnesium, titanium, and niobium, the energy gap decreased from $E_{g}$= 4.630 eV to 1.250 eV. The same phenomenon was observed for chitosan coated with Mg, Ti, and Nb, for which the bandgap changed from 4.720 eV pristine chitosan to the lowest value of 1.570 eV. These results suggest that the decrease in the cellulose and chitosan bandgap in the coated system could be related to a charge transfer from the transition metal to the cellulose and chitosan. The same phenomenon was previously observed by Mahmood et al. [42,43,44,45,46]. A similar phenomenon was observed in copper-decorated, nitrogen-doped defective graphene nanoribbons, in which the presence of copper decreased the bandgap from 3.399 eV to 3.352 eV [47].

### 3.1. Interaction of Hydrogen with Pristine Cellulose and Chitosan

To test the hydrogen storage capacity of cellulose and chitosan, we first investigated the interaction of clean cellulose and chitosan with hydrogen. The strength of the interaction of H_2_ with cellulose and chitosan was measured using the following equation:(3)$$E_{ads}=E_{\left(S\right)}+E_{\left(H2\right)}-E_{\left(S+H2\right)}$$
where $E_{\left(S\right)}$ is the total energy of the substrate (cellulose or chitosan), $E_{\left(H2\right)}$ is the total energy of isolated hydrogen, and $E_{\left(S+H2\right)}$ is the total interaction energy of cellulose or chitosan with H_2_.

The interaction of hydrogen with cellulose and chitosan was determined using Equation (3); the equilibrium parameters are summarized in Table 3.

The results presented in Table 2 reveal a weak interaction (physisorption) between hydrogen and cellulose and between H_2_ and chitosan. The calculated binding energy between H_2_ and cellulose is 0.095 eV, compared to 0.090 eV for chitosan. These results align with results previously reported in the literature [48]. Furthermore, by increasing the number of H_2_ molecules in pure cellulose and chitosan to two, a decrease in the binding energy is observed: $E_{b}$ = 0.083 eV and $E_{b}$ = 0.050 eV, for cellulose and chitosan, respectively. The optimized structures are displayed in Figure 4.

These results show that pure cellulose and pure chitosan are not favorable storage media alone. Therefore, to increase their adsorption capacity, we doped them with magnesium, titanium, and niobium; these results are described in the following section.

### 3.2. Hydrogen Storage of Cellulose Coated with Magnesium, Niobium, and Titanium

Our hydrogen interaction results with pure cellulose and chitosan show that these new materials are not efficient for hydrogen storage under ambient conditions. Therefore, to enhance the hydrogen storage capacity, we functionalized the materials to increase the active site of hydrogen adsorption. We investigated hydrogen storage on cellulose functionalized with magnesium, niobium, and titanium. The optimized structures are depicted in Figure 5.

The strength of the binding between hydrogen and cellulose was calculated as follows:(4)$$E_{ads}=\frac{E_{\left(Cel+TMs\right)}+E_{\left(nH2\right)}-E_{\left(\left(Cel+TMs\right)+nH2\right)}}{n}$$
where $E_{\left(Cel+TMs\right)}$ is the total energy of cellulose doped with transition metals (TMs = Mg, Nb, and Ti); $E_{\left(nH2\right)}$ is the total energy of nH_2_ molecules; and $E_{\left(\left(Cel+TMs\right)+nH2\right)}$ is the total energy of cellulose doped with Mg, Nb, and Ti with n H_2_ molecules adsorbed on its surface, where *n* indicates the number of adsorbed H_2_ molecules. The adsorption energy was computed using Equation (4); the equilibrium parameters are summarized in Table 4.

Cellulose coated with Nb atoms can store six H_2_ molecules in the quasi-molecular form with a distance between H atoms in the range of 0.789 to 0.896 Å and a corresponding binding distance in the range of 1.900 to 2.250 Å. Their corresponding adsorption energy fluctuates in the range of 0.198–0.765 eV before reaching saturation, as shown in Table 4. To better visualize the successive adsorption of hydrogen, Figure 6a shows the variation in hydrogen binding energy with niobium-doped cellulose with varying numbers of adsorbed hydrogen molecules. The change in adsorption energy with the number of hydrogen molecules in titanium-doped cellulose is presented in Figure 6b. Figure 6a shows that the binding energy decreases as the number of added H_2_ molecules increases. However, in cellulose doped with titanium, the maximum storage capacity is five H_2_ molecules in the quasi-molecular form, and the corresponding equilibrium parameters are d_H-H = [0.792 to 0.858 Å] and d_TM-H = [1.895–2.231 Å]. Furthermore, their corresponding binding energy varies in the range of 0.120 eV to 0.640 eV. Magnesium-doped cellulose is not displayed in Figure 4 because its storage capacity is poor. It can only adsorb two H_2_ molecules with the following equilibrium parameters: d_H-H= [0.757–0.758 Å], d_TM-H = [3.050–3.125 Å], and binding energy in the range of 0.086–0.112 eV, as shown in Table 3. These results for cellulose are in agreement with previously reported results reported in the literature [49].

To study the desorption process, we used the van’t Hoff equation [2,48,49], as expressed below, to evaluate the desorption temperature ($T_{D}$):(5)$$T_{D}=\left(\frac{E_{ads}}{K_{B}}\right){\left(\left(\frac{\Delta S}{R}-ln\left(\frac{P}{P_{o}}\right)\right)\right)}^{-1}$$
where $E_{ads}$ is the binding energy, *K_B_* represents Boltzmann’s constant (8.61733 × 10^−5^ eV/K), *P* denotes the pressure (reference pressure *Po* = 1 *atm*), *R* is the universal gas constant (8.314 JK^−1^mol^−1^), and Δ*S* is the entropy change as H_2_ moves from the gas to the liquid phase. Assume that *P* = 1 is the atmospheric pressure, and Δ*S* = 130.7 JK^−1^mol^−1^ [49,50].

The variation in the binding energy with respect to absorbed H_2_ in cellulose-doped titanium is displayed in Figure 6c.

We determine the desorption temperature for the successive addition of H_2_ on cellulose doped with niobium, titanium, and magnesium using Equation (5); the results are summarized in Table 4. Cellulose-doped niobium and titanium are the most favorable for hydrogen storage, as shown in Table 4. Therefore, to better understand the correlation between the desorption temperature (*T_D_*) and the adsorption energy, Figure 6c,d shows the desorption temperature as a function of the binding energy for the two most favorable composites (cellulose doped with niobium and cellulose doped with titanium). Figure 6c,d shows a linear relationship between the desorption temperature and the binding energy. The temperature variation in titanium-doped cellulose is expressed as by *T_D_* = 1277 × E_b_ + 0.55. In niobium-doped cellulose, the temperature varies: *T_D_* = 1277 × E_b_ − 0.071. The desorption temperature of the successive hydrogen addition in niobium-doped cellulose is in the interval of 253–978 K, where the maximum temperature is (T_D_ = 978 K) to release all the adsorbed hydrogen at a atmospheric pressure of 1. However, in the case of titanium-doped cellulose, the desorption temperature is within the range of 153–818 K, where T_D_ = 818 K is the maximum temperature required to release all the adsorbed hydrogen at the standard pressure.

### 3.3. Hydrogen Storage on Chitosan Coated with Magnesium, Niobium, and Titanium

A comparative study was also performed on chitosan doped with magnesium, niobium, and titanium as a hydrogen storage medium. The optimized structure of the chitosan doped with magnesium, titanium, and niobium with the maximum H_2_ storage capacity is shown in Figure 7. The first H_2_ molecule dissociation is noticeable in titanium- and niobium-doped chitosan, with a binding energy of 0.615 eV for niobium-doped chitosan (Ch_Nb) and 0.405 eV for titanium-doped chitosan (Ch_Ti). However, in magnesium-doped chitosan (Ch_Mg), the first Ch_Mg adsorption energy is 0.142 eV.

The successive adsorption energy of hydrogen with chitosan functionalized with the metals was determined using Equation (3). The calculation results for the subsequent H_2_ additions are summarized in Table 5.

Table 5 shows that H_2_ interacted with magnesium-coated chitosan in the gas phase, with an adsorption energy of 0.142 eV for the first adsorbed H_2_ molecule, decreasing to 0.078 eV for the second H_2_ molecule. In the case of magnesium-coated cellulose, the first adsorption energy of the first H_2_ molecule is $\;E_{b}$ = 0.112 eV and 0.086 eV for the second adsorption. Moreover, the corresponding equilibrium parameters of chitosan are as follow: critical distance (d_H-Mg): 3.141 Å for the first H_2_ molecule and 3.304 Å for the last added H_2_ molecule. The results can be compared to the equilibrium parameters for magnesium-doped cellulose, where d_H-Mg = 3.050 Å for the first adsorption and d_H-Mg = 3.125 Å for the second adsorption. The desorption temperature (*T_D_*) for these two successive adsorptions is *T_D_* = 181 K and *T_D_* = 99 K, respectively, compared with $T_{D}$= 143 K and $T_{D}$ = 109 K in magnesium-doped cellulose.

To better, observe the correlation between the adsorption energy and the number of added H_2_ molecules. The maximum hydrogen storage capacity for niobium-coated chitosan is five hydrogen molecules, as shown in Figure 8, and the addition of the six H_2_ molecules is not stable. This result indicates that niobium-doped chitosan can store five H_2_ molecules in the quasi-molecular form before reaching saturation. Figure 8a,c shows the binding energy for the successive H_2_ additions. The desorption temperature (*T_D_*) of successive H_2_ adsorption varies in the energy interval of 136–786 K. However, in the case of titanium-doped chitosan, the maximum storage capacity is three H_2_ molecules in the quasi-molecular form, with binding energy in the range of 0.132–0.405 eV. A fourth H_2_ molecule is in an unstable configuration, with an adsorption energy of 0.099 eV. The corresponding desorption temperature for the successive addition of molecules varies in the range of 168–517 K. To better understand the correlation between the adsorption energy and the corresponding desorption temperature, Figure 8c shows the desorption temperature changes (*T_D_*) with respect to the hydrogen binding energy for niobium-doped chitosan. Figure 8d also displays the desorption temperature (*T_D_*) for different hydrogen adsorption energy for titanium-doped chitosan. Furthermore, the relation between the desorption temperature (*T_D_*) and the binding energy (E_b_) is expressed by *T_D_* = 1290 × E_b_ − 1.46 in niobium-doped chitosan and T_D_ = 1280 × E_b_ − 0.3 in the case of titanium-doped chitosan.

## 4. Conclusions

In summary, the present study of hydrogen storage in pure cellulose and chitosan, as well as cellulose and chitosan doped with magnesium, titanium, and niobium using density functional theory highlights the mechanism of hydrogen storage. The results of this investigation show that the interaction between hydrogen and pure cellulose and chitosan takes place in the gas phase (physisorption). Additionally cellulose and chitosan coated with magnesium, titanium, and niobium show exciting results. Our calculations predict that cellulose doped with niobium is the most favorable medium, with a storage capacity of six H_2_ molecules, adsorption energy in the range of 0.198–0.765 eV, with system release of all the hydrogen at 978 K. Niobium-doped chitosan can accommodate five H_2_ molecules, with binding energy in the range of 0.107–0.615 eV, whereas titanium-doped cellulose has a storage capacity of four H_2_ molecules, with binding energy in the range of 0.120–0.640 eV and a maximum desorption temperature of T_max_ = 818 K, compared to a storage capacity of four H_2_ molecules in the case of titanium-doped chitosan, with an adsorption range of 0.099–0.405 eV and a maximum desorption temperature of T_max_ = 517 K. However, magnesium-doped cellulose and chitosan show an insufficient hydrogen storage capacity of two H_2_ molecules physisorbed. These results demonstrate that niobium-doped cellulose and chitosan might play an important role in the search for efficient and inexpensive hydrogen storage media.

## Figures and Tables

**Figure 1 materials-15-07573-f001:**
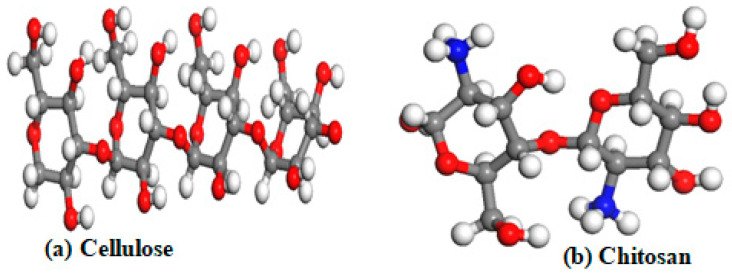
Optimized structures of pure cellulose and pristine chitosan, where red atoms are O, grey atoms are carbon (C), blue atoms are N, and white atoms are H.

**Figure 2 materials-15-07573-f002:**
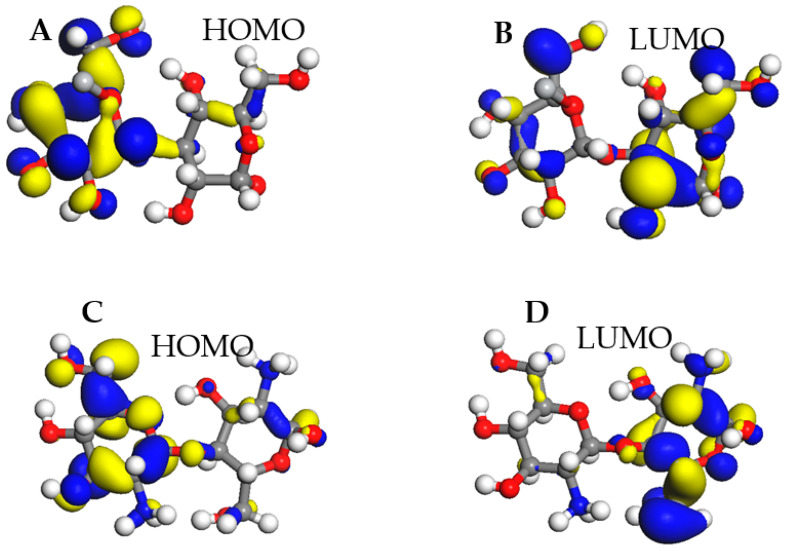
Frontier molecular orbitals, (**A**,**C**) stand for HOMO for cellulose and chitosan respectively, and (**B**,**D**) stand for LUMO for cellulose and chitosan respectively, the chemical hardness (α), ionization energy (I), electron affinity (A) of pristine cellulose and chitosan.

**Figure 3 materials-15-07573-f003:**
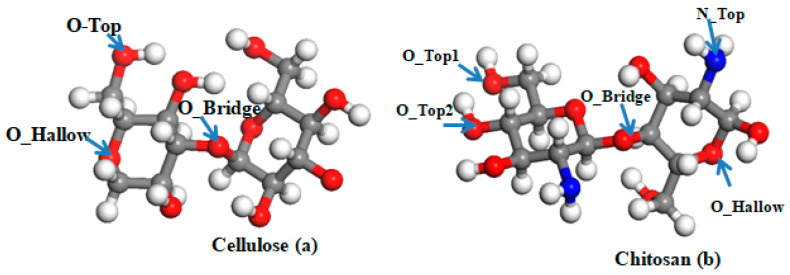
The adsorption sites (O_Top, O_Hallow, O_Bridge, N_Top, O_Top1, and O_Top2) of cellulose and chitosan.

**Figure 4 materials-15-07573-f004:**
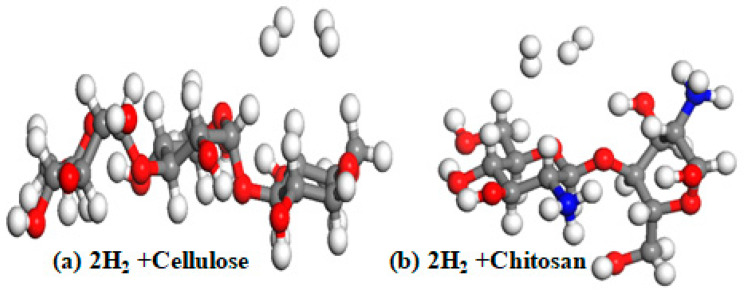
Optimized pure cellulose structure (**a**) and pure chitosan structure (**b**); red atoms are O atom, grey atoms are carbon (C), blue atoms are N, and white atoms are H.

**Figure 5 materials-15-07573-f005:**
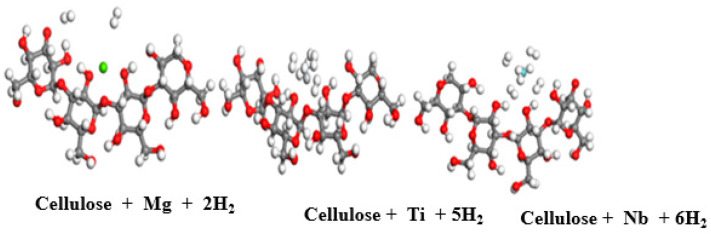
Optimized structure of magnesium-, titanium-, and niobium-doped cellulose with successive addition hydrogen molecules; red atoms are O, grey atoms are carbon (C), green atoms are Mg, grey metallic atoms are Ti, forest green atoms are Nb, and white atoms are H.

**Figure 6 materials-15-07573-f006:**
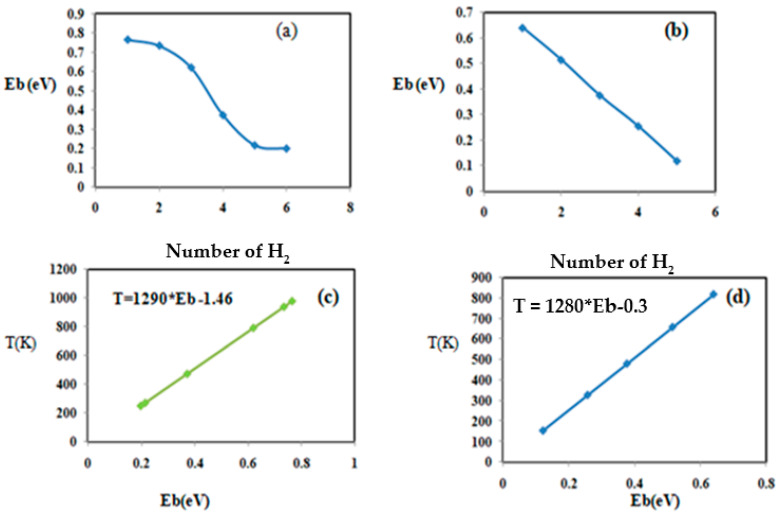
Physical and chemical parameters of the successive addition of H_2_ on cellulose. Case (**a**) represents the variation in the binding energy with respect to the number of hydrogen molecules (cellulose + Nb), case (**b**) represents the variation in the binding energy with respect to the number of H_2_ molecules (cellulose + Ti), case (**c**) represents the variation in the desorption temperature with respect to the adsorption energy (cellulose + Nb), and case (**d**) represents the variation in the desorption temperature with respect the adsorption energy of H_2_ (cellulose + Ti).

**Figure 7 materials-15-07573-f007:**
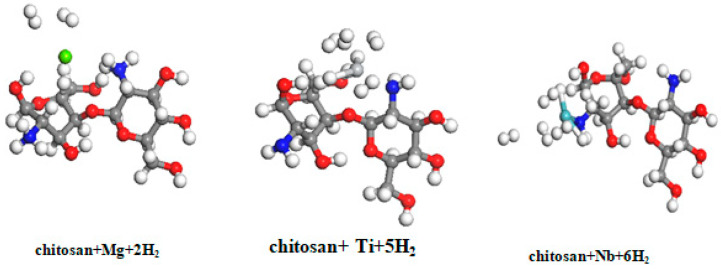
Optimized structure of the interaction of magnesium-, titanium-, and niobium-doped chitosan with the successive addition of hydrogen molecules; red atoms are O, blue atoms are N, grey atoms are carbon (C), green atoms are Mg, grey metallic atoms are Ti, forest green atoms are Nb, and white atoms are H.

**Figure 8 materials-15-07573-f008:**
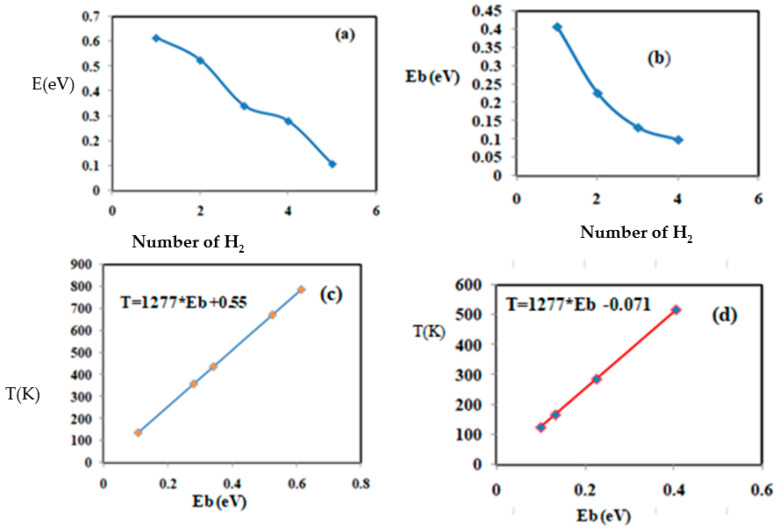
Physical and chemical parameters of the successive addition of H_2_ to chitosan. Case (**a**) represents the variation in the binding energy with respect to the number of hydrogen molecules (chitosan + Nb), case (**b**) represents the variation in the binding energy with respect to the number of H_2_ molecules (chitosan + Ti), case (**c**) represents the variation in the desorption temperature with respect to the adsorption energy (chitosan + Nb), and case (**d**) represents the variation in the desorption temperature with respect to the adsorption energy of H_2_ (chitosan + Ti).

**Table 1 materials-15-07573-t001:** Chemical parameters of cellulose and chitosan: ionization energy, electron affinity, energy gap, chemical potential, chemical hardness, chemical softness, and maximum charge transfer values.

Cellulose Parameters	
$E_{HOMO}$	−16.899
$E_{LUMO}$	−12.269
Ionization energy (I)	16.899
Electron affinity (A)	12.269
Energy gap (Eg)	4.630
Electronegativity (α)	14.584
Chemical potential (µ)	−14.584
Chemical hardness (β)	2.315
Chemical softness (S)	0.216
Maximum charge transfer (Dn)	3.149
Chitosan parameters	
$\;E_{HOMO}$	−5.918
$E_{LUMO}$	−1.196
Ionization energy (I)	5.918
Electron affinity (A)	1.196
Energy gap (Eg)	4.721
Electronegativity (α)	3.557
Chemical potential (µ)	−3.557
Chemical hardness (β)	2.361
Chemical softness (S)	0.212
Maximum charge transfer (Dn)	1.507

**Table 2 materials-15-07573-t002:** Equilibrium parameters (binding energy ($E_{b}$) and critical distance (d_F (Å))) and the bandgap ($E_{g}$ (eV)) of the interaction between pristine cellulose and chitosan with magnesium, titanium, and niobium.

Substrate	*E_b_* (eV)	d_F (Å)	*E_g_* (eV)
cellulose + Mg	1.890	2.544	2.46
cellulose + Ti	3.72	2.340	1.50
cellulose + Nb	3.726	2.3	0.25
pristine cellulose			4.630
pristine chitosan			4.720
chitosan + Mg	1.547	3.666	2.770
chitosan +Ti	5.450	3.255	1.570
chitosan + Nb	8.185	2.307	2.980

**Table 3 materials-15-07573-t003:** Enthalpy energy of H_2_ with pure cellulose and chitosan.

Cellulose + H_2_	0.095
Cellulose + 2H_2_	0.083
Chitosan + H_2_	0.090
Chitosan +2H_2_	0.050

**Table 4 materials-15-07573-t004:** Equilibrium parameters: binding energy ($E_{b}$), critical distance (d_TM-H (Å)), distance between hydrogen atoms, and desorption temperature (T_D._) between cellulose doped with magnesium, titanium, and niobium.

Number of H_2_ Molecules	$E_{b}$ (eV)	d_H-H (Å)	d_TM-H (Å)	T_D_ (K)
Cell + H_2_ + Nb	0.765	0.896	1.900	978
	0.734	0.897	1.910	938
	0.620	0.870	1.996	792
	0.372	0.840	1.999	475
	0.215	0.831	2.020	274
	0.198	0.789	2.250	253
Cell + Ti + H_2_	0.640	0.850	1.920	818
	0.515	0.858	1.895	658
	0.376	0.831	1.997	480
	0.256	0.807	2.010	327
	0.120	0.792	2.231	153
Cell + Mg + H_2_	0.112	0.758	3.050	143
	0.086	0.757	3.125	109

**Table 5 materials-15-07573-t005:** Successive hydrogen adsorption on chitosan coated with magnesium, titanium, and niobium, along with their corresponding binding distance (d_H-M), the distance between hydrogen atoms (d_H-H), and the desorption temperature (*T_D_*).

Number of H_2_ Molecules	Binding Energy	d_H-H (Å)	d_H-M (Å)	*T_D_* (K)
Ch + Mg + H_2_	0.142	0.752	3.141	99
	0.078	0.761	3.304	181
	0.101	0.756	3.280	129
Ch + Ti +H_2_	0.405	0.794	2.025	517
2H_2_	0.225	0.790	2.068	287
	0.132	0.799	2.011	168
	0.099	0.787	2.076	126
Ch + Nb + H_2_				
	0.615	0.816	2.012	786
	0.525	0.825	1.995	671
	0.341	0.823	2.020	436
	0.280	0.835	2.045	358
	0.107	0.800	2.890	136

## Data Availability

Not applicable.

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
