# Peer review of "A Comparative Density Functional Theory Study of Hydrogen Storage in Cellulose and Chitosan Functionalized by Transition Metals (Ti, Mg, and Nb)"

_materials, 2022, doi:10.3390/ma15217573_

Round 1

Reviewer 1 Report

Omar Faye et al. reported that the hydrogen storage in pure cellulose, chitosan and cellulose, and chitosan doped with magnesium, titanium, and niobium is analyzed using density functional theory. This study demonstrated that hydrogen and pure cellulose, and chitosan interaction occurs in the gas phase. The cellulose and chitosan-doped with magnesium titanium and niobium show exciting results. Our calculations predict that cellulose doped with niobium is the most favorable medium where 6H2 molecules are stored. Compared with 5H2 molecules stored in niobium doped chitosan with Tmax=818 K to release all H2 molecules.

I recommend this manuscript for publishing in the “Materials” with the following improvements.

1.     Improve the language of the abstract overall and try to write an effective abstract which may attract the attention of readers.

2.     Modify abstract portion into past tense.

3.     Author should must mention the level of theory utilized in DFT calculations, in the abstract portion.

4.     Add band gap findings in the Abstract part.

5.     Always use a definite article before the name of any specific theory or technique, etc. You should use “The” before density functional theory in the first line of the abstract.

6.     A line in the introduction part “Also, a minimum cost ($/kg H2) of 266” is known as a fragment which is not a suitable way to write an article. Adjust it in some other sentence.

7.     Traditional symbol for binding energy is Eb. But in the manuscript it is written as Eb. Author should correct it throughout the manuscript.

8.     Author should plot graphs in the Figure 5 again. There exists no symmetry, even symbol for biding energy is not correct and units are not italicized.

9.     There must be a one-line space between a quantity and its unit. Also, the unit must be in italics format. This space and format are missing in your manuscript in most of the times. For example, “5.5wt% and 40g/L” must be written with a properly as “5.5 wt% and 40 g/L” in the introduction part.

10.  The word et al. must be used in italics format. Correct it in the whole manuscript.

11.  A lot of grammatical mistakes are observed in the manuscript such as “Therefore, to have a smooth transition from fossil fuel to clean and sustainable fuel.” must be written as “Therefore, a smooth transition from fossil fuel to clean and sustainable fuel is required.” The language of manuscript should always be meaningful.

12.  The equations of your manuscript seems to be distorted. Author should rewrite them through some other source.

13.  The grids of tables must be removed throughout your manuscript. Tables format is not appealing.

14.  In the Introduction part, in the line # 50 and 51, “[-40, 60], [-40, 85]” are written to mention physical parameters, but they are looking like as they are also the citations of references. Use some different way to write them.

15.  The format of Figure 7 needs to be corrected. Maintain uniformity in labels of figures and tables of your manuscript.

16.  A space should be introduced before unit and units must be italicized as [0.198 – 0.765 eV].

17.  In common practice, Figures and Tables first letter should be capital as “Equation3”. This mistakes are repeated many time in the manuscript.

18.  The graphical interface must be improved in its resolution as well as in detail for creating a significant impact of the manuscript.

19.  Add some information about DFT in the Introduction part. Take help from following articles and also cite in the manuscript,

doi.org10.1007/s10876-019-01573-0; doi.org/10.5012/bkcs.2014.35.5.1391

doi.org/10.1098/rsos.210570; doi.org/10.1016/j.arabjc.2021.103295

doi.org/10.1039/D1RA00876E

20.  In the sentence, “Where Eads … from the literature [39,40]” a lot of mistakes are found and it should be written as “Where Eads is the binding energy, KB represents Boltzmann's constant (8.61733×10−5 eV /K), P denotes the pressure, the reference pressure Po = 1 atm, R is the universal gas constant (8.314 JK−1mol−1). ΔS is the entropy change as H2 moves from gas to the liquid phase. Taking P=1 atm pressure and bringing the value of ΔS = 130.7 J K−1mol−1 from the literature [39,40]”.

21.  I suggest to add Frontier Molecular Orbital Analysis into a separate discussion section in order to make it convenient to understand. In order to improve its discussion study and cite the following articles,

doi.org/10.1016/j.arabjc.2014.11.007; doi.org/10.1002/bkcs.10526; doi.org/10.1016/j.comptc.2020.112797; doi.org/10.1016/j.comptc.2021.113387; doi.org/10.1002/poc.3427

22.  In addition to band gap calculation, I suggest to calculate global softness and hardness via computing Global Reactivity Parameters, for investigated systems to discuss their reactivity. Take help from following articles and cite in write-up,

23.  Correct the sentence “Furthermore, the energy band gap was determined b using equation 2” as “Furthermore, the energy band gap was determined by using Equation 2”.

Author Response

Response to Reviewer Comments on Manuscript Number:  materials-1930399

Title: A comparative density functional theory study of hydrogen storage in cellulose and chitosan functionalized by transition metals (Ti, Mg, and Nb).

Article Type: Full-Length Article

 Journal of materials

Reviewer # 1

Comments and Suggestions for Authors

Omar Faye et al. reported that the hydrogen storage in pure cellulose, chitosan, cellulose, and chitosan doped with magnesium, titanium, and niobium is analyzed using density functional theory. This study demonstrated that hydrogen and pure cellulose, and chitosan interaction occurs in the gas phase. The cellulose and chitosan-doped with magnesium titanium and niobium show exciting results. Our calculations predict that cellulose doped with niobium is the most favorable medium where 6H2 molecules are stored. Compared with 5H2 molecules stored in niobium doped chitosan with Tmax=818 K to release all H2 molecules.

I recommend this manuscript for publishing in the “Materials” with the following improvements.

  1. Improve the language of the abstract overall and try to write an effective abstract which may attract the attention of readers.

The authors reply: We have rewritten the overall abstract

  1. Modify abstract portion into past tense.

The author's reply: The abstract has been modified and verbs are in the past tense

  1. Author should must mention the level of theory utilized in DFT calculations, in the abstract portion.

The authors reply: We thank the reviewer for his comment; we have added the DFT theory calculations in the abstract.

  1. Add band gap findings in the Abstract part.

The authors reply: We thank the reviewer, for his remark, we have added the band gap in the abstract.

  1. Always use a definite article before the name of any specific theory or technique, etc. You should use “The” before density functional theory in the first line of the abstract.

The author's reply: We thank the reviewer for his remark; we have added the definite article before the name of any theory

  1. A line in the introduction part “Also, a minimum cost ($/kg H2) of 266” is known as a fragment which is not a suitable way to write an article. Adjust it in some other sentence.

The authors reply: We have corrected the sentence.  

  1. Traditional symbol for binding energy is Eb. But in the manuscript it is written as Eb. Author should correct it throughout the manuscript.

The authors reply: We have corrected all the binding energy symbol in the manuscript.

  1. Author should plot graphs in the Figure 5 again. There exists no symmetry, even symbol for biding energy is not correct and units are not italicized.

The authors reply: We thank the reviewer for his remark, we have plotted again in Figure 5.

  1. There must be a one-line space between a quantity and its unit. Also, the unit must be in italics format. This space and format are missing in your manuscript in most of the times. For example, “5.5wt% and 40g/L” must be written with a properly as “5.5 wt% and 40 g/L” in the introduction part.

The authors reply: We thank the reviewer for his comments, we have corrected all unit formats in the manuscript

  1. The word et al. must be used in italics format. Correct it in the whole manuscript.

The authors reply: We thank the reviewer for his comment, we have corrected the comment

  1. A lot of grammatical mistakes are observed in the manuscript such as “Therefore, to have a smooth transition from fossil fuel to clean and sustainable fuel.” must be written as “Therefore, a smooth transition from fossil fuel to clean and sustainable fuel is required.” The language of manuscript should always be meaningful.

The authors reply: We have read and corrected the grammatical mistakes throughout the manuscript.

  1. The equations of your manuscript seems to be distorted. Author should rewrite them through some other source.

The authors reply: We thank the reviewer for his remark, all equations have been rewritten

  1. The grids of tables must be removed throughout your manuscript. Tables format is not appealing.

The authors reply: We thank the reviewer for his comment, all grids of the table are removed.

  1. In the Introduction part, in the line # 50 and 51, “[-40, 60], [-40, 85]” are written to mention physical parameters, but they are looking like as they are also the citations of references. Use some different ways to write them.

The authors reply: We thank the reviewer for his remark; changes have been made to differentiate them from citations.

  1. The format of Figure 7 needs to be corrected. Maintain uniformity in the labels of figures and tables of your manuscript.

The authors reply: We thank the reviewer for his comments, we have corrected Figure 7

  1. A space should be introduced before the unit and units must be italicized as [0.198 – 0.765 eV].

The authors reply: We thank the reviewer for his remark, space has been introduced before the unit and is italicized

  1. In common practice, Figures and Tables first letter should be capital as “Equation3”. This mistakes are repeated many time in the manuscript.

The authors reply: We thank the reviewer for his comment, Figures and Tables first letter are capitalized

  1. The graphical interface must be improved in its resolution as well as in detail for creating a significant impact of the manuscript.

The authors reply: We thank the reviewer; we have made a new graphical interface figure again.

  1. Add some information about DFT in the Introduction part. Take help from following articles and also cite in the manuscript,

doi.org10.1007/s10876-019-01573-0; doi.org/10.5012/bkcs.2014.35.5.1391                   

doi.org/10.1098/rsos.210570; doi.org/10.1016/j.arabjc.2021.103295

doi.org/10.1039/D1RA00876E

The authors reply: We thank the reviewer for his suggestion, we have added the DFT information in the introduction by citing the mentioned papers.

  1. In the sentence, “Where Eadsfrom the literature [39,40]” a lot of mistakes are found and it should be written as “Where Eads is the binding energy, KB represents Boltzmann's constant (8.61733×10−5 eV /K), P denotes the pressure, the reference pressure P= 1 atm, R is the universal gas constant (8.314 JK−1mol−1). ΔS is the entropy change as H2 moves from gas to the liquid phase. Taking P=1 atm pressure and bringing the value of ΔS = 130.7 J K−1mol−1 from the literature [39,40]”.

The authors reply: We thank, the reviewer for his comment, and we have corrected the sentence.

  1. I suggest adding Frontier Molecular Orbital Analysis into a separate discussion section in order to make it convenient to understand. In order to improve its discussion study and cite the following articles,

doi.org/10.1016/j.arabjc.2014.11.007; doi.org/10.1002/bkcs.10526; doi.org/10.1016/j.comptc.2020.112797; doi.org/10.1016/j.comptc.2021.113387; doi.org/10.1002/poc.3427

The authors reply: We thank the reviewer for his suggestion; we have added the mentioned paper.

  1. In addition to band gap calculation, I suggest to calculate global softness and hardness viacomputing Global Reactivity Parameters, for investigated systems to discuss their reactivity. Take help from following articles and cite in write-up.

The author's reply: We thank the reviewer for his suggestion; we have calculated global softness and hardness 

  1. Correct the sentence “Furthermore, the energy band gap was determined b using equation 2” as “Furthermore, the energy band gap was determined by using Equation 2”.

The authors reply: We thank the reviewer for his comment, and we have corrected the sentence.

Response to Reviewer Comments on Manuscript Number:  materials-1930399

Title: A comparative density functional theory study of hydrogen storage in cellulose and chitosan functionalized by transition metals (Ti, Mg, and Nb).

Article Type: Full-Length Article

 Journal of materials

Reviewer # 1

Comments and Suggestions for Authors

Omar Faye et al. reported that the hydrogen storage in pure cellulose, chitosan, cellulose, and chitosan doped with magnesium, titanium, and niobium is analyzed using density functional theory. This study demonstrated that hydrogen and pure cellulose, and chitosan interaction occurs in the gas phase. The cellulose and chitosan-doped with magnesium titanium and niobium show exciting results. Our calculations predict that cellulose doped with niobium is the most favorable medium where 6H2 molecules are stored. Compared with 5H2 molecules stored in niobium doped chitosan with Tmax=818 K to release all H2 molecules.

I recommend this manuscript for publishing in the “Materials” with the following improvements.

  1. Improve the language of the abstract overall and try to write an effective abstract which may attract the attention of readers.

The authors reply: We have rewritten the overall abstract

  1. Modify abstract portion into past tense.

The author's reply: The abstract has been modified and verbs are in the past tense

  1. Author should must mention the level of theory utilized in DFT calculations, in the abstract portion.

The authors reply: We thank the reviewer for his comment; we have added the DFT theory calculations in the abstract.

  1. Add band gap findings in the Abstract part.

The authors reply: We thank the reviewer, for his remark, we have added the band gap in the abstract.

  1. Always use a definite article before the name of any specific theory or technique, etc. You should use “The” before density functional theory in the first line of the abstract.

The author's reply: We thank the reviewer for his remark; we have added the definite article before the name of any theory

  1. A line in the introduction part “Also, a minimum cost ($/kg H2) of 266” is known as a fragment which is not a suitable way to write an article. Adjust it in some other sentence.

The authors reply: We have corrected the sentence.  

  1. Traditional symbol for binding energy is Eb. But in the manuscript it is written as Eb. Author should correct it throughout the manuscript.

The authors reply: We have corrected all the binding energy symbol in the manuscript.

  1. Author should plot graphs in the Figure 5 again. There exists no symmetry, even symbol for biding energy is not correct and units are not italicized.

The authors reply: We thank the reviewer for his remark, we have plotted again in Figure 5.

  1. There must be a one-line space between a quantity and its unit. Also, the unit must be in italics format. This space and format are missing in your manuscript in most of the times. For example, “5.5wt% and 40g/L” must be written with a properly as “5.5 wt% and 40 g/L” in the introduction part.

The authors reply: We thank the reviewer for his comments, we have corrected all unit formats in the manuscript

  1. The word et al. must be used in italics format. Correct it in the whole manuscript.

The authors reply: We thank the reviewer for his comment, we have corrected the comment

  1. A lot of grammatical mistakes are observed in the manuscript such as “Therefore, to have a smooth transition from fossil fuel to clean and sustainable fuel.” must be written as “Therefore, a smooth transition from fossil fuel to clean and sustainable fuel is required.” The language of manuscript should always be meaningful.

The authors reply: We have read and corrected the grammatical mistakes throughout the manuscript.

  1. The equations of your manuscript seems to be distorted. Author should rewrite them through some other source.

The authors reply: We thank the reviewer for his remark, all equations have been rewritten

  1. The grids of tables must be removed throughout your manuscript. Tables format is not appealing.

The authors reply: We thank the reviewer for his comment, all grids of the table are removed.

  1. In the Introduction part, in the line # 50 and 51, “[-40, 60], [-40, 85]” are written to mention physical parameters, but they are looking like as they are also the citations of references. Use some different ways to write them.

The authors reply: We thank the reviewer for his remark; changes have been made to differentiate them from citations.

  1. The format of Figure 7 needs to be corrected. Maintain uniformity in the labels of figures and tables of your manuscript.

The authors reply: We thank the reviewer for his comments, we have corrected Figure 7

  1. A space should be introduced before the unit and units must be italicized as [0.198 – 0.765 eV].

The authors reply: We thank the reviewer for his remark, space has been introduced before the unit and is italicized

  1. In common practice, Figures and Tables first letter should be capital as “Equation3”. This mistakes are repeated many time in the manuscript.

The authors reply: We thank the reviewer for his comment, Figures and Tables first letter are capitalized

  1. The graphical interface must be improved in its resolution as well as in detail for creating a significant impact of the manuscript.

The authors reply: We thank the reviewer; we have made a new graphical interface figure again.

  1. Add some information about DFT in the Introduction part. Take help from following articles and also cite in the manuscript,

doi.org10.1007/s10876-019-01573-0; doi.org/10.5012/bkcs.2014.35.5.1391                   

doi.org/10.1098/rsos.210570; doi.org/10.1016/j.arabjc.2021.103295

doi.org/10.1039/D1RA00876E

The authors reply: We thank the reviewer for his suggestion, we have added the DFT information in the introduction by citing the mentioned papers.

  1. In the sentence, “Where Eadsfrom the literature [39,40]” a lot of mistakes are found and it should be written as “Where Eads is the binding energy, KB represents Boltzmann's constant (8.61733×10−5 eV /K), P denotes the pressure, the reference pressure P= 1 atm, R is the universal gas constant (8.314 JK−1mol−1). ΔS is the entropy change as H2 moves from gas to the liquid phase. Taking P=1 atm pressure and bringing the value of ΔS = 130.7 J K−1mol−1 from the literature [39,40]”.

The authors reply: We thank, the reviewer for his comment, and we have corrected the sentence.

  1. I suggest adding Frontier Molecular Orbital Analysis into a separate discussion section in order to make it convenient to understand. In order to improve its discussion study and cite the following articles,

doi.org/10.1016/j.arabjc.2014.11.007; doi.org/10.1002/bkcs.10526; doi.org/10.1016/j.comptc.2020.112797; doi.org/10.1016/j.comptc.2021.113387; doi.org/10.1002/poc.3427

The authors reply: We thank the reviewer for his suggestion; we have added the mentioned paper.

  1. In addition to band gap calculation, I suggest to calculate global softness and hardness viacomputing Global Reactivity Parameters, for investigated systems to discuss their reactivity. Take help from following articles and cite in write-up.

The author's reply: We thank the reviewer for his suggestion; we have calculated global softness and hardness 

  1. Correct the sentence “Furthermore, the energy band gap was determined b using equation 2” as “Furthermore, the energy band gap was determined by using Equation 2”.

The authors reply: We thank the reviewer for his comment, and we have corrected the sentence.

Reviewer 2 Report

In this work, the authors used density functional theory to calculate the binding energy of H2 molecules on pure and metal-doped cellulose and chitosan to evaluate their hydrogen storage ability. The manuscript however suffers from methodology flaws and serious inconsistency. In its present form, I cannot accept it for publication in Materials. Below are more comments I have for this manuscript:

1.The authors claimed “no study on hydrogen storage in pure cellulose … was done” (Line 84–86), but Ref. 36 studied cellulose as a hydrogen storage material. They should change the claim to reflect the current status correctly.

2. Why do structures of cellulose differ so much between Figure 1 and 2?

3. Band gap (equation 2) should be calculated as: ELUMO–EHOMO. How can band gap values “show the ability to donate electrons to the additive atoms”, as the authors claimed in Line 127–128?

4. The authors discussed the calculated binding energies and critical distances in the text, but they are inconsistent with the values in Tables, and values are different when the same quantity appears twice. For example, binding energy of Ti on cellulose is 3.752 eV in Line 133, but it is 3.720 eV in Line 141. The authors must make sure which values are correct and make them consistent across the whole manuscript.

5. Figure 5 and 7 should be exchanged.

6. The authors plotted the desorption temperature vs. adsorption energy based on equation 5. Then why are there intercept terms in Figure 5c,d and Figure 7c,d, and the slopes are different for Figure 7c,d?

7. The authors did not discuss what adsorption sites they considered for the doped metal atoms as well as the H2 molecules. They should include this information in the manuscript.

8. The authors computed average binding energy (equation 4) to determine if multiple H2 molecules could be adsorbed. This is however not correct. For Nb-doped cellulose and chitosan, the fourth and third H2 adsorption respectively is already not favored as the adsorption is endothermic. Therefore, the storage capacity of Nb-doped cellulose and chitosan is 3H2 and 2H2 rather than 6H2 and 5H2, respectively. The capacity is thus not high, especially heavy metals are used.

Author Response

Response to Reviewer Comments on Manuscript Number:  materials-1930399

Title: A comparative density functional theory study of hydrogen storage in cellulose and chitosan functionalized by transition metals (Ti, Mg, and Nb).

Article Type: Full-Length Article

 Journal of materials

Reviewer # 2

Comments and Suggestions for Authors

In this work, the authors used density functional theory to calculate the binding energy of H2 molecules on pure and metal-doped cellulose and chitosan to evaluate their hydrogen storage ability. The manuscript however suffers from methodology flaws and serious inconsistency. In its present form, I cannot accept it for publication in Materials. Below are more comments I have for this manuscript:

  1. The authors claimed “no study on hydrogen storage in pure cellulose … was done” (Line 84–86), but Ref. 36 studied cellulose as a hydrogen storage material. They should change the claim to reflect the current status correctly.

The authors reply: We thank the reviewer for his comment, we have changed the claim to reflect the current status correctly.

  1. Why do structures of cellulose differ so much between Figure 1 and 2?

The authors reply: We thank the reviewer for his comment; the difference is only in the number of monomers

  1. Band gap (equation 2) should be calculated as: ELUMO–EHOMO. How can band gap values “show the ability to donate electrons to the additive atoms”, as the authors claimed in Line 127–128?

The authors reply: We thank the reviewer for his comment we have corrected the mistake the equation and how the band gap value can show the ability to donate electrons to the additive atoms.

  1. The authors discussed the calculated binding energies and critical distances in the text, but they are inconsistent with the values in Tables, and values are different when the same quantity appears twice. For example, binding energy of Ti on cellulose is 3.752 eV in Line 133, but it is 3.720 eV in Line 141. The authors must make sure which values are correct and make them consistent across the whole manuscript.

The authors reply: We thank the reviewer for his comment, we have corrected  the binding energy of Ti on cellulose is 3.752 eV in Line 133, but it is 3.720 eV in Line 141.

  1. Figure 5 and 7 should be exchanged.

The authors reply: We thank the reviewer for his comment, we have exchanged Figures 5 and 7.

  1. The authors plotted the desorption temperature vs. adsorption energy based on equation 5. Then why are there intercept terms in Figure 5c,d and Figure 7c,d, and the slopes are different for Figure 7c,d?

The author's reply: We thank the reviewer for his comments, the slopes are different because we are dealing with two different systems.

  1. The authors did not discuss what adsorption sites they considered for the doped metal atoms as well as the H2 molecules. They should include this information in the manuscript.

The authors reply: We thank the reviewer for his comment; we have indicated the adsorption site where the doped metal atoms have been placed on cellulose and chitosan.

  1. The authors computed average binding energy (equation 4) to determine if multiple H2 molecules could be adsorbed. This is however not correct. For Nb-doped cellulose and chitosan, the fourth and third H2 adsorption respectively is already not favored as the adsorption is endothermic. Therefore, the storage capacity of Nb-doped cellulose and chitosan is 3H2 and 2H2 rather than 6H2 and 5H2, respectively. The capacity is thus not high, especially since heavy metals are used.

The authors reply: We thank the reviewer for his comment we have checked again the storage capacity of Nb-doped cellulose and chitosan and we find that the storage of cellulose coated with niobium is  6H2 same for titanium doped cellulose is 5H2. Since saturation is reached when the binding energy goes down and the distance between hydrogen atom decrease to its initial bond length.

Reviewer 3 Report

The authors of the manuscript show a theoretical approach to the search and discovery of new materials for hydrogen storage. This field is particularly important due to the decarbonization processes that are necessary to take care of the environment.

Below I will detail some of the comments that must be answered so that this manuscript can be published.

1) Authors use the Dmol3 module for the DFT study, however, they do not reference the computational suit (like Material Studio, ...) or computational program where this module is implemented.

2)They also do not comment on which energy minimizer is used.

3)The presentation of the Mathematical formulas should be improved. In my case, they look like blurred images. I don't think it's pleasant for the readers of this to see such blurry formulas.

4) I suppose the authors have carried out a Basis Set Superposition Error BSSE) study to verify their result is optimal.

5) It would be interesting for readers of the manuscript to know with what program the Homo-Lumo study was carried out (if it was with the Dmol3 module, indicate the instruction)

6) The authors compare the number of H2 molecules that cellulose and chitosan can store depending on the metal they are doped with, however, I think it would be interesting to know how many molecules could interact with the metal without cellulose or chitosan.

7) The authors have considered how the structure of chitosan and cellulose can affect the processes presented.

Author Response

Response to Reviewer Comments on Manuscript Number:  materials-1930399

Title: A comparative density functional theory study of hydrogen storage in cellulose and chitosan functionalized by transition metals (Ti, Mg, and Nb).

Article Type: Full-Length Article

 Journal of materials

Reviewer # 3

Comments and Suggestions for Authors

The authors of the manuscript show a theoretical approach to the search and discovery of new materials for hydrogen storage. This field is particularly important due to the decarbonization processes that are necessary to take care of the environment.

Below I will detail some of the comments that must be answered so that this manuscript can be published.

1) Authors use the Dmol3 module for the DFT study, however, they do not reference the computational suit (like Material Studio, ) or computational program where this module is implemented.

The authors reply: We thank the reviewer for his comment; we have introduced the computational program where this module is implemented.

2)They also do not comment on which energy minimizer is used.         

The authors reply: We thank the reviewer for his comment, we have indicated the energy minimize used in our calculation.

3)The presentation of the Mathematical formulas should be improved. In my case, they look like blurred images. I don't think it's pleasant for the readers of this to see such blurry formulas.

The authors reply: We thank the reviewer for his comment, We have brought some changes in the mathematical formula

4) I suppose the authors have carried out a Basis Set Superposition Error BSSE) study to verify their result is optimal.

The authors reply: We thank the reviewer for his comments; we have included the Basis Set Superposition Error BSSE during our relaxation calculation.

5) It would be interesting for readers of the manuscript to know with what program the Homo-Lumo study was carried out (if it was with the Dmol3 module, indicate the instruction).

The authors reply: We thank the reviewer for his comment; we have carried the Homo and Lumo by using Dmol3.

6) The authors compare the number of H2 molecules that cellulose and chitosan can store depending on the metal they are doped with, however, I think it would be interesting to know how many molecules could interact with the metal without cellulose or chitosan.

The authors reply: We thank the reviewer for his comment; this study was already in these papers:

Faye, O., & Szpunar, J. A. (2018). An efficient way to suppress the competition between adsorption of H2 and desorption of n H2–Nb complex from graphene sheet: a promising approach to H2 storage. The Journal of Physical Chemistry C122(50), 28506-28517.

Faye, Omar, et al. "Tailoring the capability of carbon nitride (C3N) nanosheets toward hydrogen storage upon light transition metal decoration." Nanotechnology 30.7 (2018): 075404.

7) The authors have considered how the structure of chitosan and cellulose can affect the processes presented.

The authors reply: We thank the reviewer for his comment; we have considered how the structure of chitosan and cellulose can affect hydrogen storage. We know that the active site of hydrogen is the metal site therefore; the maximum hydrogen storage is around the metal site.

Round 2

Reviewer 2 Report

The authors did not address my main concerns on the methodology and inconsistency of the study in the revised manuscript. Here, I reinstate some of my comments and hope the authors could address them to improve the manuscript. In the current form, the manuscript is not suitable for publication.

1. What is the point of showing different numbers of monomers in Figure 1 and 2?

2. I highly suggest the authors check and compare all the numbers in the text with those in the tables. Many of them are inconsistent!

3. If Figure 5c,d and Figure 7c,d are based on equation 5, then there should be no intercept terms, and the slopes should be identical between c and d plots.

4. When calculating adsorption energies, different adsorption sites are usually needed to be compared to determine the best adsorption site. What adsorption sites had the authors considered for the doped metal atoms and H2 on cellulose and chitosan? Randomly choosing one site to study adsorption is not sufficient.

5. If the authors compute the adsorption energy for each H2 (one at a time consecutively) on Nb-doped cellulose, then the 4th H2 adsorption would have positive reaction energy (E(Nb@cellulose+4H2) - E(Nb@cellulose+3H2) - E(H2)). Similarly, the 3rd H2 adsorption on Nb-doped chitosan is also endothermic. Therefore, the storage capacity of Nb-doped cellulose and chitosan is only 3H2 and 2H2 rather than 6H2 and 5H2, respectively.

Author Response

Response to Reviewer Comments on Manuscript Number:  materials-1930399

Title: A comparative density functional theory study of hydrogen storage in cellulose and chitosan functionalized by transition metals (Ti, Mg, and Nb).

Article Type: Full-Length Article

 Journal of materials

I want to thank you for your time and meaningful comments. I appreciate the professionalism of the editor, I for your work too.

We have carefully read through and corrected all the queries/ comments suggested by the reviewer; please see our responses below:

Comments and Suggestions for Authors

The authors did not address my main concerns on the methodology and inconsistency of the study in the revised manuscript. Here, I reinstate some of my comments and hope the authors could address them to improve the manuscript. In the current form, the manuscript is not suitable for publication.

  1. What is the point of showing different numbers of monomers in Figure 1 and 2?

Authors reply: We thank the reviewer for his comments, the point of showing two numbers of monomers in Figures 1 and 2 is related to the fact that the. Frontier molecular orbitalsare bulk properties therefore do not depends on the size of the structure that why to gain more time we chose to use to reduce the chain of cellulose.

  1. I highly suggest the authors check and compare all the numbers in the text with those in the tables. Many of them are inconsistent!

Authors reply: We thank the reviewer for his remark; we have read the whole manuscript and corrected the inconsistency values in the main text.

  1. If Figure 5c, d and Figure 7c, d are based on equation 5, then there should be no intercept terms, and the slopes should be identical between c and d plots.

Author reply:We thank the reviewer for his remark; by extrapolation, we have an intercept. The equation is determine by choosing two points since it is a linear equation.

  1. When calculating adsorption energies, different adsorption sites are usually needed to be compared to determine the best adsorption site. What adsorption sites had the authors considered for the doped metal atoms and H2 on cellulose and chitosan? Randomly choosing one site to study adsorption is not sufficient.

Authors reply: We thank the reviewer for his comments. As showed in the Figure below, cellulose can have seven carbon asymmetric with two oxygen in different states. We know that the metal always try to bind to more free active site, since oxygen and nitrogen are more electronegativethan carbon therefore, we do not need to test all these site. It is only important if we have a homogeneous structure (same atom in the structure). The same remark is also applicable to chitosan.

  1. If the authors compute the adsorption energy for each H2 (one at a time consecutively) on Nb-doped cellulose, then the 4th H2 adsorption would have positive reaction energy (E(Nb@cellulose+4H2) - E(Nb@cellulose+3H2) - E(H2)). Similarly, the 3rd H2 adsorption on Nb-doped chitosan is also endothermic. Therefore, the storage capacity of Nb-doped cellulose and chitosan is only 3H2 and 2H2 rather than 6H2 and 5H2, respectively.

Authors reply: We thank the reviewer for his comments, we have indicated the equation to compute the succesive adsorption energy. Pleased we invite the reviewer to check these papers papers regarding the equation used to compute the succesive adsorption energy. The storage capacity is corrected

  1. Faye, Omar, and Jerzy A. Szpunar. "An efficient way to suppress the competition between adsorption of H2 and desorption of n H2–Nb complex from graphene sheet: a promising approach to H2 storage." The Journal of Physical Chemistry C50 (2018): 28506-28517.

  1. Ramos-Castillo, C. M., et al. "Palladium clusters supported on graphene monovacancies for hydrogen storage." The Journal of Physical Chemistry C15 (2015): 8402-8409.

  1. Liu, Yang, et al. "Hydrogen storage using Na-decorated graphyne and its boron nitride analog." International journal of hydrogen energy24 (2014): 12757-12764.

Response to Reviewer Comments on Manuscript Number:  materials-1930399

Title: A comparative density functional theory study of hydrogen storage in cellulose and chitosan functionalized by transition metals (Ti, Mg, and Nb).

Article Type: Full-Length Article

 Journal of materials

I want to thank you for your time and meaningful comments. I appreciate the professionalism of the editor, I for your work too.

We have carefully read through and corrected all the queries/ comments suggested by the reviewer; please see our responses below:

Comments and Suggestions for Authors

The authors did not address my main concerns on the methodology and inconsistency of the study in the revised manuscript. Here, I reinstate some of my comments and hope the authors could address them to improve the manuscript. In the current form, the manuscript is not suitable for publication.

  1. What is the point of showing different numbers of monomers in Figure 1 and 2?

Authors reply: We thank the reviewer for his comments, the point of showing two numbers of monomers in Figures 1 and 2 is related to the fact that the. Frontier molecular orbitalsare bulk properties therefore do not depends on the size of the structure that why to gain more time we chose to use to reduce the chain of cellulose.

  1. I highly suggest the authors check and compare all the numbers in the text with those in the tables. Many of them are inconsistent!

Authors reply: We thank the reviewer for his remark; we have read the whole manuscript and corrected the inconsistency values in the main text.

  1. If Figure 5c, d and Figure 7c, d are based on equation 5, then there should be no intercept terms, and the slopes should be identical between c and d plots.

Author reply:We thank the reviewer for his remark; by extrapolation, we have an intercept. The equation is determine by choosing two points since it is a linear equation.

  1. When calculating adsorption energies, different adsorption sites are usually needed to be compared to determine the best adsorption site. What adsorption sites had the authors considered for the doped metal atoms and H2 on cellulose and chitosan? Randomly choosing one site to study adsorption is not sufficient.

Authors reply: We thank the reviewer for his comments. As showed in the Figure below, cellulose can have seven carbon asymmetric with two oxygen in different states. We know that the metal always try to bind to more free active site, since oxygen and nitrogen are more electronegativethan carbon therefore, we do not need to test all these site. It is only important if we have a homogeneous structure (same atom in the structure). The same remark is also applicable to chitosan.

  1. If the authors compute the adsorption energy for each H2 (one at a time consecutively) on Nb-doped cellulose, then the 4th H2 adsorption would have positive reaction energy (E(Nb@cellulose+4H2) - E(Nb@cellulose+3H2) - E(H2)). Similarly, the 3rd H2 adsorption on Nb-doped chitosan is also endothermic. Therefore, the storage capacity of Nb-doped cellulose and chitosan is only 3H2 and 2H2 rather than 6H2 and 5H2, respectively.

Authors reply: We thank the reviewer for his comments, we have indicated the equation to compute the succesive adsorption energy. Pleased we invite the reviewer to check these papers papers regarding the equation used to compute the succesive adsorption energy. The storage capacity is corrected

  1. Faye, Omar, and Jerzy A. Szpunar. "An efficient way to suppress the competition between adsorption of H2 and desorption of n H2–Nb complex from graphene sheet: a promising approach to H2 storage." The Journal of Physical Chemistry C50 (2018): 28506-28517.

  1. Ramos-Castillo, C. M., et al. "Palladium clusters supported on graphene monovacancies for hydrogen storage." The Journal of Physical Chemistry C15 (2015): 8402-8409.

  1. Liu, Yang, et al. "Hydrogen storage using Na-decorated graphyne and its boron nitride analog." International journal of hydrogen energy24 (2014): 12757-12764.